# Next-Generation Sequencing for Determining the Effect of Arginine on Human Dental Biofilms Using an In Situ Model

**DOI:** 10.3390/pharmacy9010018

**Published:** 2021-01-12

**Authors:** Nanako Kuriki, Yoko Asahi, Maki Sotozono, Hiroyuki Machi, Yuichiro Noiri, Mikako Hayashi, Shigeyuki Ebisu

**Affiliations:** 1Department of Restorative Dentistry and Endodontology, Osaka University Graduate School of Dentistry, Suita, Osaka 565-0871, Japan; yoko-a@dent.osaka-u.ac.jp (Y.A.); m.sotozono@dent.osaka-u.ac.jp (M.S.); mikarin@dent.osaka-u.ac.jp (M.H.); ebisu@dent.osaka-u.ac.jp (S.E.); 2Osaka University Dental Technology Institute, Suita, Osaka 565-0871, Japan; machi@dent.osaka-u.ac.jp; 3Department of Oral Health Science, Division of Cariology, Operative Dentistry and Endodontics, Niigata University Graduate School of Medical and Dental Sciences, Niigata 951-8514, Japan; noiri@dent.niigata-u.ac.jp

**Keywords:** dental biofilm, next-generation sequencing, biofilm control, in situ model, arginine

## Abstract

Oral biofilms are associated with caries, periodontal diseases, and systemic diseases. Generally, antimicrobial therapy is used as the first line of treatment for infectious diseases; however, bacteria in biofilms eventually develop antibiotic resistance. This study aimed to apply our in situ biofilm model to verify whether an arginine preparation is useful for plaque control. Ten healthy subjects who did not show signs of caries, gingivitis, or periodontitis were recruited. The dental biofilms from the subjects were obtained using our oral device before and after gargling with arginine solution for 4 weeks. We found that 8% arginine solution significantly increased the concentration of ammonium ions (NH_4_^+^) in vitro and in vivo in saliva (*p* < 0.05) and decreased the proportions of the genera *Atopobium* and *Catonella* in vivo. However, the viable count was unaffected by the mouthwash. Further, oral populations of the genera *Streptococcus* and *Neisseria* tended to increase with the use of arginine. Therefore, we concluded that using an 8% arginine solution decreased the NH_4_^+^ concentration in the oral cavity without affecting the number of viable bacteria, and that the diversity of oral bacterial flora changed. We suggest that arginine might help prevent mature biofilm formation.

## 1. Introduction

More than 700 species of bacteria inhabit the human oral cavity, and these bacteria form biofilms [1,2]. Not only bacteria, but also fungi, viruses, and parasites are present in biofilms. The genus *Candida* is often detected as a fungus in the oral cavity, and it has been reported that its prevalence is increasing, especially in denture wearers and the elderly, and this can lead to invasive infections with a high mortality rate [3]. In addition, periodontal pockets are rich in nutrients, and parasites such as *Trichomonas* have been detected, which exacerbate damage to the oral mucosa and have been detected in oral ulcers in kidney transplant patients [4]. Oral biofilms are formed in an ordered way and retain a diverse microbial composition that remains relatively stable over time in good health [5]. However, imbalances in the oral bacterial flora (dysbiosis) caused by specific stress factors, such as carbohydrate consumption and plaque accumulation, are thought to lead to the development of oral diseases such as caries and periodontal disease [6,7]. In caries, there is a shift to community domination by acid-producing and acid-resistant species such as *Streptococcus mutans* and *Lactobacillus* [8]. Periodontal disease is thought to require colonization by certain pathogens such as *Porphyromonas gingivalis* [9]. Among them, it is considered that the synergistic and antagonistic actions among the bacterial species in the biofilm enhance or inhibit the colonization and pathogenicity of each [10,11]. Thus, oral biofilms might lead to the development of oral diseases, and the interactions among its constituent bacteria affect bacterial virulence [12]. Therefore, studying the mechanism underlying the formation and the control/suppression of dental biofilms is of clinical significance to prevent and treat oral diseases and improve general health.

Unlike biofilms formed in vitro, dental biofilms formed in the human oral cavity are affected by diverse oral environmental and host factors, such as host immunity, pH, enzymes, saliva, and antibiotic use [13]. Therefore, it is extremely important to develop an in situ model for the formation and evaluation of dental biofilms in the human oral cavity. We previously developed an oral device that enables the formation and quantitative evaluation of dental biofilms in the human oral cavity over a period and confirmed that the number of viable bacteria in supragingival biofilms increased in two steps [14]. Furthermore, using next-generation sequencing (NGS) to analyze the human oral microbiome in great detail, we found that the initial bacterial flora, comprising facultative anaerobic bacteria, is replaced by a bacterial flora comprising gram-negative anaerobic bacteria during oral biofilm formation, indicating that the proportion of anaerobic bacteria that cause gingivitis, such as *Fusobacterium*, *Prevotella*, and *Porphyromonas* [15,16], increased after 48 h, suggesting a shift to mature biofilms. Therefore, to suppress the growth of such mature biofilms, their mechanical and chemical removal with brushing and mouthwashes are routinely performed. However, a scientifically based method for controlling dental biofilms has not yet been established.

In recent years, the number of elderly people has been increasing worldwide, and the risk of developing caries and periodontal disease is increasing. This is due to the deterioration of oral hygiene and the decrease in saliva, which causes acidification of the oral environment and the accumulation of plaque, resulting in imbalances of oral bacterial flora [6,7]. In addition, the interrelationship between systemic diseases such as diabetes and heart disease and periodontal disease has been reported [17]. Therefore, the importance of controlling dental biofilms is increasing. Several in vitro studies have been conducted on the control/suppression of dental biofilms using antibacterial agents and quorum-sensing-related substances [18,19]. However, bacteria in biofilms develop resistance to antibiotics, resulting in growth inhibition of bacterial flora that do not form biofilms [20]. Therefore, the approach of converting biofilm-forming bacterial flora to flora that does not form biofilms has been attracting attention worldwide.

Arginine, one of the components of human saliva, has been the focus of attention for altering the oral microbial community from pathogenic bacteria-dominated flora that causes dental caries and periodontal disease to nonpathogenic bacterial flora [21]. The effects of arginine on oral bacteria have been studied in vitro. It is reported that exogenous L-arginine inhibits the attachment of *Streptococcus mutans* to saliva-treated substrates, promotes *Streptococcus gordonii* single-species biofilm formation, and increases H_2_ O_2_ generation in biofilms [22]. Arginine is metabolized by oral biofilms, primarily by a 3-enzyme pathway called the arginine deiminase system (ADS) [21]. ADS is a system in which non-acid-resistant bacteria, such as *Streptococcus sanguinis*, *S. gordonii*, and *Streptococcus parasanguinis*, produce ammonium ions and raise the pH to protect the bacteria themselves. It has been reported that this stabilizes the oral flora and prevents it from changing to a carious environment using an in vitro model [23]. Additionally, a peptide of *Streptococcus cristatus* (Arc A), which is involved in the ADS, suppresses the expression of Fim A in *Porphyromonas gingivalis* [24]. A clinical epidemiology study showed that the saliva concentration of free arginine in caries-free individuals is significantly higher than that in caries-affected individuals [25]. It has also been reported that using a dentifrice containing 1.5% arginine reduces the occurrence of dental caries, compared to that with the use of a dentifrice containing only fluorine [26]. However, there are no reports of studies quantitatively evaluating the effect of arginine on dental biofilms formed in the human oral cavity. Therefore, the aim of this study was to verify whether an arginine preparation is useful for plaque control using our in situ biofilm model.

## 2. Materials and Methods

### 2.1. Determining the Effective Arginine Concentration In Vitro

An in vitro study was conducted to determine the effective concentration of arginine mouthwash used in the in situ studies. Biofilms obtained from saliva samples were prepared according to a previous report [27]. In detail, the saliva collected from three subjects was cultured aerobically in a 96-well plate at 37 °C for 4 h. Arginine solution at various concentrations (0, 0.8, 1.6, 2.4, 3.2, 4.0, 4.8, 5.6, 6.4, 7.2, and 8.0%) was added to the wells for 1 min. Sterile distilled water was used in the control group. Adenosine triphosphate (ATP) bioluminescence assay for quantifying the metabolic activity of biofilms was performed according to a previous study [28]. Briefly, at the end of the batch test, a biofilm sample was taken and transferred to a 1.5-mL tube. The sample was centrifuged at 5000 rpm for 10 min, and a lysis reagent was added. Luciferase assay was performed by adding purified luciferase enzyme (CellTiter-Glo^®^ Luminescent Cell Viability Assay, Promega, Tokyo, Japan) to a standard solution (10 mM DDL4 in 50 mM tris-HCl (pH 7.8), 25 °C, and 2 mM luciferin). Light emission was measured using a 1420 Multilabel counter ARVO™ MX (PerkinElmer Japan Co., Ltd., Kanagawa, Japan) and expressed as relative light units. Bradford assay was used to determine the protein content of the sample. Ammonium ion (NH_4_^+^) concentration in the remaining 10 μL of the bacterial solution was measured using a coulometric titration type ammonia meter AT-2000 (Central Science, Tokyo, Japan) to evaluate arginine activity.

### 2.2. In Situ Study

#### 2.2.1. Study Subjects

The in situ study recruited 10 healthy participants (six men and four women) aged 26–37 years (mean, 30.4 ± 3.2 years) who were students and staff members of Osaka University Graduate School of Dentistry during the time of study. We defined healthy subjects based on criteria previously reported [29]. Clinical or radiological signs of caries, gingivitis, or periodontitis were not detected in any of the subjects. The subjects did not undergo any antibiotic treatment 6 months prior to inclusion in the study.

#### 2.2.2. In Situ Model of Biofilms

Dental biofilms were prepared using an in situ biofilm model previously developed by the principal investigators [14]. Samples of the dental biofilm were obtained using an oral device developed by the authors. The oral device was attached to the upper jaw similarly to a night guard, and hydroxyapatite (HA) disks (6 mm diameter, 1.5 mm height) were attached to the buccal side of the molars. The subjects wore this device during sleep for 8 h.

#### 2.2.3. Experimental Protocol for Extracting Biofilms

The experimental protocol is shown in Figure 1. Subjects of the control group were equipped with the oral device during sleep from 12 a.m. to 8 a.m., and the HA disks were collected immediately after waking up. For 4 weeks, subjects of the arginine group gargled using a mouthwash containing the aforementioned effective concentration of arginine. Similarly to the control group, the oral device was then worn from 12 a.m. to 8 a.m., and the disks were collected immediately after waking up. After formation of the dental biofilms, the disks were extracted without disrupting the adherent biofilm, as previously described [30]. In addition, saliva samples were collected at the same time point for measuring NH_4_^+^ concentrations as described in Section 2.1. Resting saliva was collected, and 10 μL was used for the experiment.

#### 2.2.4. Determination of Viable Counts

The sample was immersed in sterile distilled water, stirred for 5 min, sonicated for 30 s, and then continuously diluted 10-fold. The diluted bacterial solution was seeded on Columbia sheep blood agar medium (Becton, Dickinson and Company, Fukushima, Japan) at 37 °C for 48 h aerobically and anaerobically using an Aneropack^®^, and the viable cell count was measured. Each sample was tested in triplicate.

#### 2.2.5. DNA Sequencing

DNA was extracted from the collected sample using a DNeasy^®^ PowerSoil DNA Isolation Kit (QIAGEN, Hilden, Germany), and the V1-V2 region of the 16S DNA was amplified using a MiSeq^®^ instrument (Illumina Inc., San Diego, CA, USA) with a universal primer targeting 16S rRNA (27F mod: ACACTCTTTCCCTACACGACGCTCTTCCGATCTNNNNN- AGRGTTTGATYMTGGCTCAG), (338R: GTGACTGGAGTTCAGACGTGTGCTCTTCCGATCTNNNNN-TGCTGCCTCCCGTAGGAGT) and sequenced according to the manufacturer’s instructions. The sequences were clustered into operational taxonomic units based on a 97% homology cutoff. We also analyzed functional predictors. The sequences were analyzed using the QIIME pipeline. The obtained sequences were systematically compared to existing sequences using the EzBioCloud 16S database. Moreover, functional predictive analyses were performed by creating a functional composition table using PICRUSt (ver.1.1.1) based on the Kyoto Encyclopedia of Genes and Genomes ortholog.

### 2.3. Statistical Analysis

Differences in ATP and NH_4_^+^ concentrations in the in vitro study were evaluated using one-way analysis of variance and the Tukey–Kramer test using SAS software version 10.02 2012 (SAS Institute Inc., Cary, NC, USA). The tests were performed using a risk rate of 5%. Significant differences in results from the in situ study were tested using SPSS^®^ Statistics software (version 22.0, IBM SPSS Inc., Chicago, IL, USA), and the data are displayed as box plots. The Wilcoxon rank sum test was performed using a risk rate of 5%.

## 3. Results

### 3.1. Effective In Vitro Arginine Concentrations

When ATP bioluminescence assays were performed at the selected arginine concentrations, no significant difference was observed in ATP levels (Figure 2). Further, NH_4_^+^ concentrations in the 8% arginine group were significantly higher than those in the control group (distilled water), indicating high arginine activity (Figure 3). Based on these results, the effective concentration of arginine was determined as 8%, which was used for the in situ studies.

### 3.2. Effect of Arginine In Situ

#### 3.2.1. Effect of Arginine on the Viable Count and NH_4_^+^ Concentration In Situ

There was no change in the viable cell count under aerobic and anaerobic conditions between the control group and the 8% arginine group (Figure 4). However, the NH_4_^+^ concentration in the saliva of subjects from the 8% arginine group was significantly higher than that in the saliva of subjects from the control group (Figure 5).

#### 3.2.2. Effects of Arginine on the Dental Biofilm Flora and Functional Factors In Situ

First, when the two groups were compared at the phylum level, the phylum Firmicutes tended to increase and the phylum Bacteroidetes tended to decrease in the arginine group compared with levels in the control group (Figure 6). At the genus level, the genera Streptococcus and Neisseria of the arginine group showed an increasing tendency (Figure 7). To determine whether these changes were significant, we restricted our analysis to the most abundant genera that represented > 0.05% of the taxa. Following this, the proportions of Atopobium and Catonella were found to have decreased significantly in the arginine group compared to those in the control group (Figure 8).

Our further analyses focused on the effects of the most frequently detected functional factors that averaged > 0.05% of the proportions. The level of ribosomal large subunit pseudouridine synthase B of the arginine group decreased significantly compared to that in the control group (Figure 9).

## 4. Discussion

In recent years, studies on the intestinal microflora have revealed that the balance in human microflora is closely related to health. In 2013, it was reported that *Fusobacterium nucleatum*, one of the periodontopathic bacteria detected in dental biofilms, might cause colon cancer [31] and that the relationship between dental biofilms and systemic diseases is of great interest. With the development of NGS, comprehensive gene analyses of bacterial flora inhabiting the human body have been performed worldwide [32,33]. It has been reported that NGS is the best analytical tool for studying the population in an oral biofilm [34]. However, quantitative analyses of the changes in bacterial flora in the human oral cavity have not been conducted to understand the process of dental biofilm formation. Therefore, it is extremely important to develop a model that can enable the formation and evaluation of dental biofilms in the human oral cavity and to study the mechanisms underlying biofilm formation, as well as control/suppression methods. Therefore, we developed an oral device that can enable the formation and quantitative evaluation of dental biofilms in the human oral cavity over a period of time. Our in situ model made it possible to analyze the diversity of the inhabiting oral bacteria [14].

Dental diseases, such as caries and periodontal disease, are currently considered to result from an imbalance (dysbiosis) in the normally stable resident oral flora [35]. This is because perturbations in the microbiome caused by certain stress factors, such as carbohydrate consumption or plaque accumulation, can lead to the development of oral diseases, such as caries or periodontal diseases [5]. Therefore, the approach of maintaining the balance of the bacterial flora by changing the pathogenic bacterial flora to nonpathogenic flora rather than killing the bacteria has attracted attention worldwide [5,36]. The use of arginine has been reported to induce the ADS and prevent the oral environment from turning acidic by increasing the pH of the oral environment, leading to the development of a nonpathogenic bacterial flora that is less likely to produce caries [21,37]. It is generally advocated that it is important to limit sugars and acids in foods and beverages to control saliva pH [6,7]. In the ADS, acid-sensitive bacteria produce NH_4_^+^ and raise the pH to protect themselves. As a result, the teeth are less likely to become carious. Ammonia produced via the ADS and protonated to NH_4_^+^ is one of the major pH-increasing factors in healthy oral environments [13]. Therefore, we measured the NH_4_^+^ concentration in this study to investigate arginine activity. First, we examined the effective concentration of arginine in an in vitro study. It was found that the amount of biofilm did not change at any concentration of arginine used (Figure 2); NH_4_^+^ concentration was the highest when an 8% arginine solution was used, among the various concentrations tested (Figure 3). Therefore, we determined 8% arginine as the effective arginine concentration.

Our in situ model was used to investigate the effect of arginine on viable cell count and arginine activity of oral dental biofilms. There was no change in viable cell count under aerobic and anaerobic conditions between the control group and the 8% arginine group (Figure 4). However, NH_4_^+^ concentration in the saliva of subjects from the 8% arginine group was significantly higher than that in the saliva of subjects from the control group (Figure 5). Thus, significant arginine activity was observed in the oral cavity of the subjects from the 8% arginine group, but the number of bacteria was not affected. This is because arginine has the ability of altering the oral microbial community from pathogenic bacteria-dominated flora that cause dental caries and periodontal disease to nonpathogenic bacterial flora, not killing the bacteria [21].

Analysis of the bacterial flora revealed individual differences in the data of healthy subjects. Although this is consistent with previous reports [14,38], statistical analyses revealed certain trends (Figure 6, Figure 7 and Figure 8). First, at the phylum level, it was found that Firmicutes increased and Bacteroidetes decreased in proportion in the arginine group, compared with those in the control group (Figure 6). At the genus level, *Streptococcus* and *Neisseria* were found to have increased in proportion (Figure 7), whereas *Atopobium* and *Catonella* decreased significantly in proportion (Figure 8). In our previous report, we found that species of *Streptococcus* and *Neisseria* are abundant in early biofilms, whereas species of *Fusobacterium*, *Porphyromonas*, and *Prevotella* are abundant in mature biofilms 48 h after formation, increased after 48 h [14]. A study that investigated the effect of the time interval between brushing one’s teeth and the induction of gum inflammation found that brushing every 48 h increases plaque score formation, although gingivitis did not develop. The authors speculated that these changes to this mature biofilm may be explained by the quantitative and qualitative changes in biofilms that occur approximately 48 h after initial growth [39]. This study found that the arginine group tended to have a higher proportion of *Streptococcus* and *Neisseria*, similar to the observation in the bacterial flora of early biofilms [14]. Therefore, it is suggested that arginine activity might prevent the transition from early biofilms to mature biofilms and prevent dysbiosis. The genera *Atopobium* and *Catonella* are present at low proportions in the oral flora, and *Atopobium* has been added to the list of cariogenic bacteria [40,41]. Moreover, *Atopobium* contains species that are rare opportunistic pathogens, even though they have been detected in skin and soft tissue infections, as well as in infective endocarditis in patients with untreated diabetes [42]. Therefore, it is suggested that arginine could have a function in preventing dysbiosis by inhibiting the growth of opportunistic pathogens, such as *Atopobium* and *Catonella*. However, these are bacteria that have been recently detected and have hardly been studied, and thus, further knowledge of these bacteria is required.

Bacteria in biofilms also communicate via quorum sensing using competence stimulating peptide (CSP), which contributes to an increase in antimicrobial resistance. Thus, oral bacteria in the biofilm do not exist as independent entities but function as a metabolically integrated microbial community [8]. Therefore, we focused on bacterial metabolism and analyzed functional predictors. We analyzed the presence of functional predictors that comprised an average of >0.05% of the proportions and found that the level of ribosomal large subunit pseudouridine synthase B of the arginine group decreased significantly compared to that in the control group (Figure 9). Pseudouridine synthase is an enzyme involved in the most frequent post-transcriptional modification of cellular RNA. Crystallographic analysis showed that all pseudouridine synthases share a common core fold and active site structure, and this core is modified by peripheral domains, accessory proteins, and guide RNAs, resulting in significant substrate diversity [43]. Pseudouridine modification has also been suggested to be involved in the development of human diseases, such as mitochondrial myopathy with lactic acidosis and sideroblastic anemia, and congenital dyskeratosis [44]. In this study, pseudouridine synthase levels were found to be reduced in the arginine group, which suggests that arginine might reduce the diversity of bacterial cell substrates and promote the stability of the oral environment.

In this study, it was predicted that the number of indigenous bacteria in the oral cavity that are not acid-resistant and do not possess arginine-related functional factors would increase in the 8% arginine group. However, in this study, we found that these bacteria and functional factors did not increase in number. In the future, it will be necessary to identify more detailed taxonomic compositions and functional factors using whole genome shotgun sequencing and RNA-seq. Moreover, we intend to develop further dental biofilm control/suppression methods by creating effective arginine preparations, such as adding drugs that enhance the effects of arginine.

## 5. Conclusions

This study investigated whether an arginine preparation was useful for biofilm control. For this purpose, our in situ model of biofilms was selected as the system for quantitative evaluation of dental biofilms in the human oral cavity. We found that the use of 8% arginine solution increased the concentration of NH_4_^+^ in the oral cavity and altered the diversity of the oral flora but did not affect the number of viable bacteria in the oral cavity. This suggests that an arginine solution might be effective as a prebiotic. Through such research, it is expected that the establishment of control/suppression methods for dental biofilms will lead to the prevention and treatment of oral diseases, and the promotion of general health.

## Figures and Tables

**Figure 1 pharmacy-09-00018-f001:**
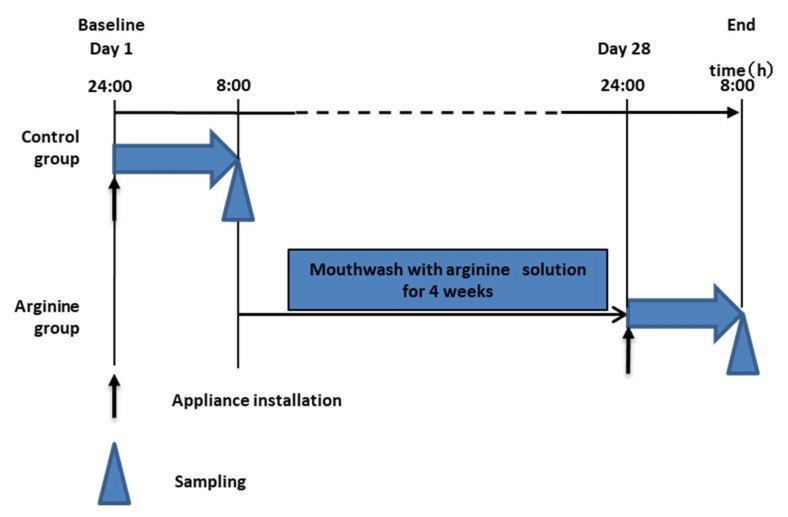
Experimental protocol for biofilm extraction from subjects of the arginine and control groups. Sampling of hydroxyapatite (HA) disks and saliva was performed according to the above experimental schedule to conduct DNA sequencing and to measure NH_4_^+^ concentrations. Black arrows indicate use of the oral device. A triangular mark indicates the sampling time point.

**Figure 2 pharmacy-09-00018-f002:**
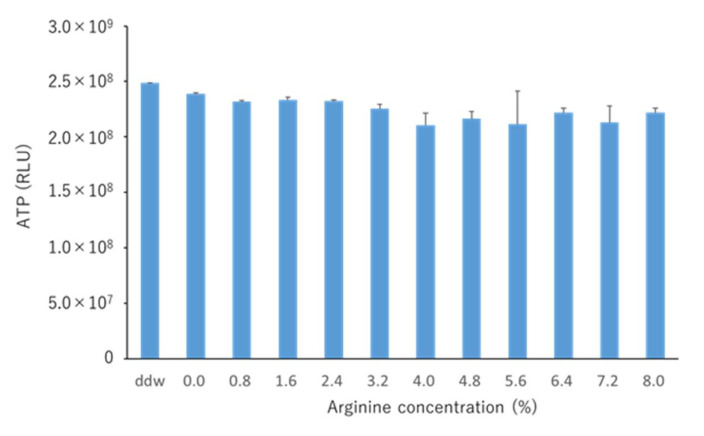
ATP bioluminescence of saliva-derived biofilms treated with different concentrations of arginine. Distilled water was used as the control. Luciferase and Bradford assays were performed for measuring bioluminescence and protein contents of the samples. Each sample was tested in triplicate. No significant difference was observed between the various arginine concentrations (one-way analysis of variance, Tukey–Kramer test, *n* = 3, *p* > 0.05). ATP, adenosine triphosphate; RLU, relative fluorescence unit; ddw, distilled water.

**Figure 3 pharmacy-09-00018-f003:**
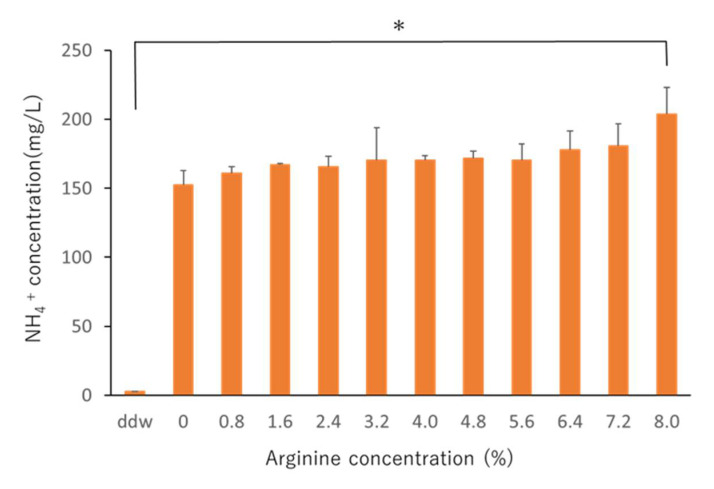
NH_4_^+^ concentrations of saliva-derived biofilms treated with different concentrations of arginine. Distilled water was used as the control. NH_4_^+^ concentrations were measured using a coulometric titration type ammonia meter AT-2000 (Central Science). Each sample was tested in triplicate. The NH_4_
^+^ concentration of the 8% arginine-treated sample was significantly higher than that of the control (one-way analysis of variance, Tukey–Kramer test, *n* = 3, * *p* < 0.05).

**Figure 4 pharmacy-09-00018-f004:**
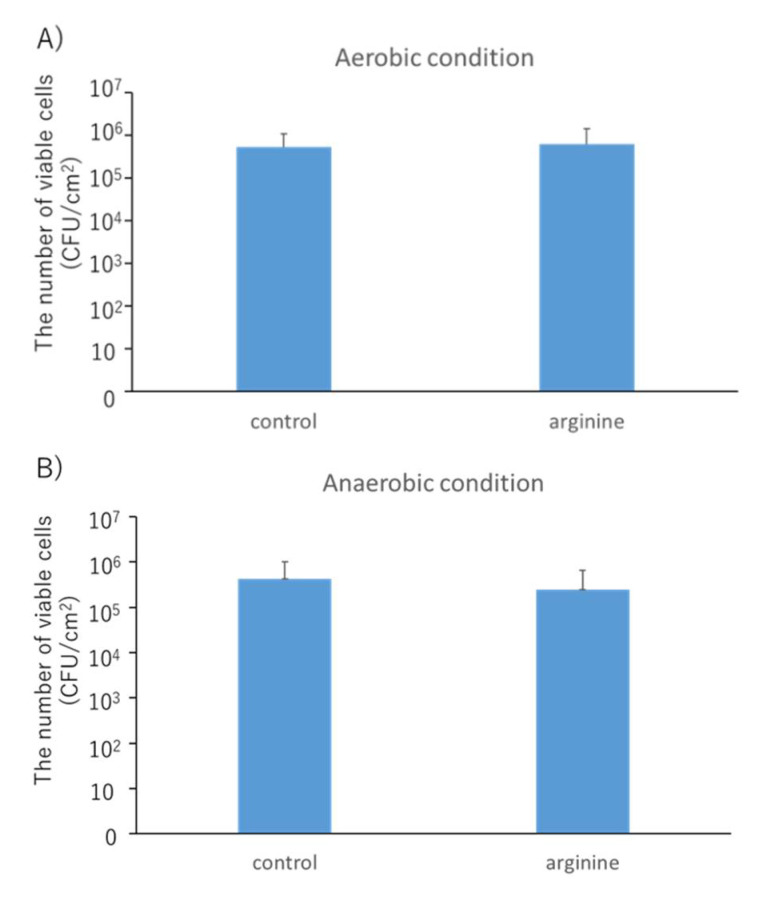
Viable bacterial cell counts of biofilms obtained from the 8% arginine and control groups under aerobic (**A**) or anaerobic (**B**) conditions. Serially diluted samples were cultured on Columbia sheep blood agar medium at 37 °C for 48 h aerobically and anaerobically using an Aneropack^®^. Each sample was tested in triplicate. No change in the viable cell count was observed under both conditions (Wilcoxon rank sum test, *n* = 10, *p* > 0.05). CFU, colony forming unit.

**Figure 5 pharmacy-09-00018-f005:**
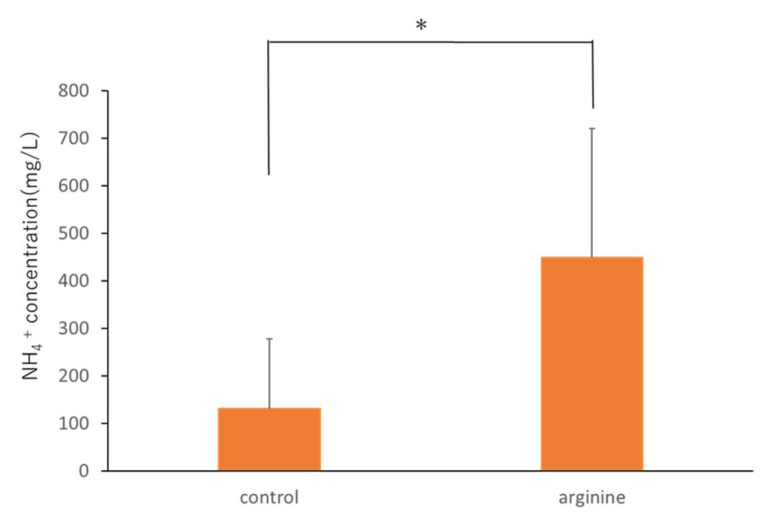
NH_4_^+^ concentrations of the biofilms obtained from the 8% arginine and control groups. NH_4_^+^ concentrations were measured using a coulometric titration type ammonia meter AT-2000 (Central Science). Each sample was tested in triplicate. Wilcoxon rank sum test, *n* = 10, * *p* < 0.05.

**Figure 6 pharmacy-09-00018-f006:**
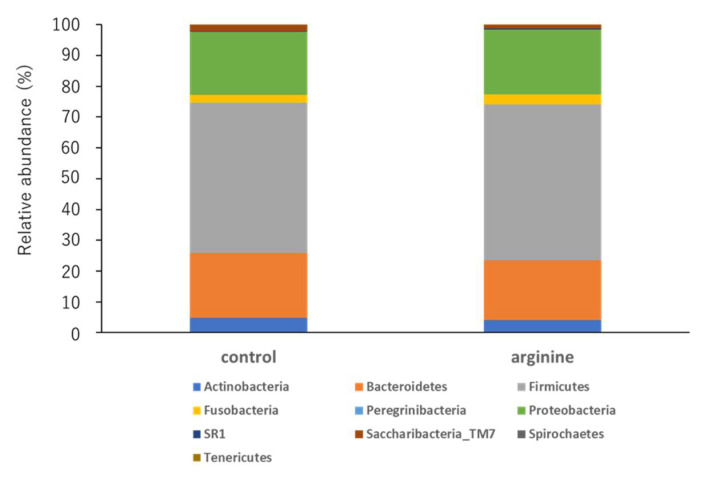
Relative abundances of bacterial phyla in all subjects (*n* = 10). The 16S rRNA obtained from the DNA of the biofilm samples was sequenced using next-generation sequencing. This graph represents the average of data collected from all subjects.

**Figure 7 pharmacy-09-00018-f007:**
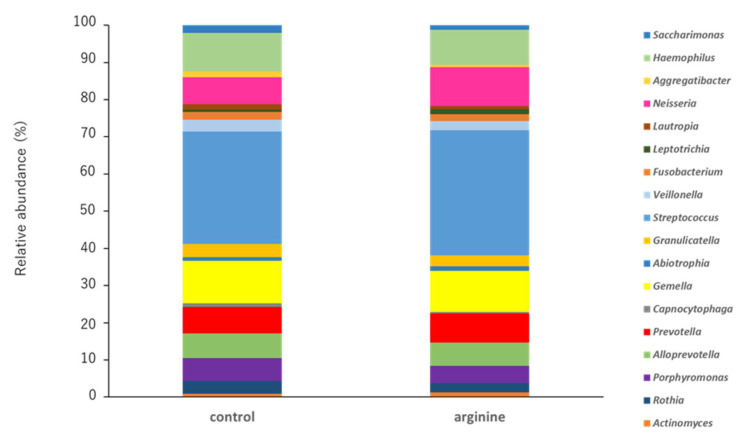
Relative abundances of bacterial genera in all subjects (*n* = 10). The 16S rRNA obtained from the DNA of the biofilm samples was sequenced using next-generation sequencing. The most frequently detected taxa (>1% relative abundance) in each level are shown.

**Figure 8 pharmacy-09-00018-f008:**
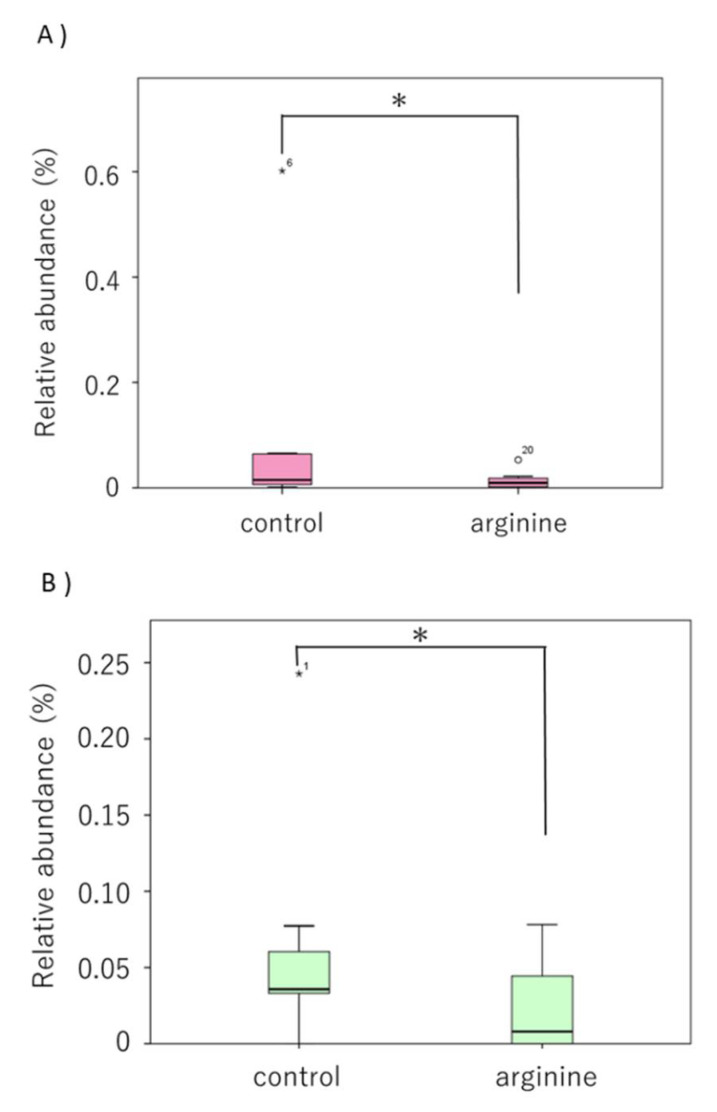
Relative proportion of bacterial genera with significant differences (**A**) *Atopobium*, (**B**) *Catonella*. Boxes extend from the 25th to 75th percentiles. Circles represent outliers. Asterisks above the whisker indicate a significant difference (Wilcoxon rank sum test, *n* = 10, * *p* < 0.05).

**Figure 9 pharmacy-09-00018-f009:**
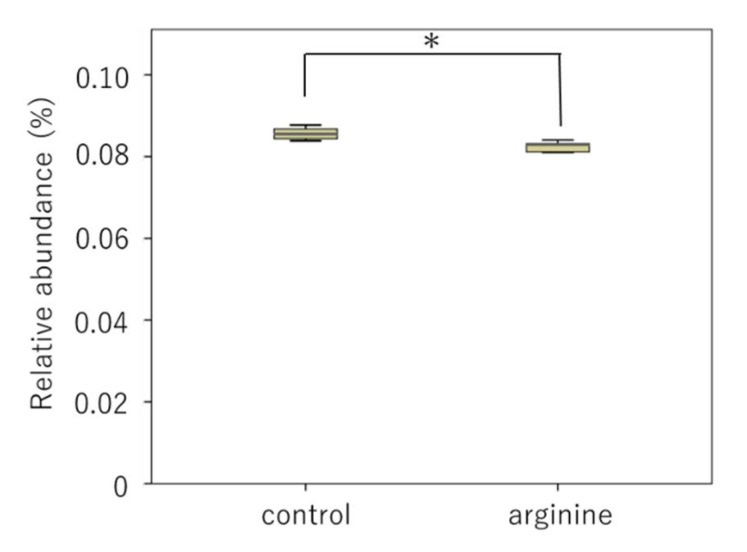
Relative proportion of the ribosomal large subunit pseudouridine synthase B with significant differences. Boxes extend from the 25th to 75th percentiles. Asterisks above the whisker indicate a significant difference (Wilcoxon rank sum test, *n* = 10, * *p* < 0.05).

## Data Availability

Data are available on request due to restrictions (e.g., privacy or ethical). Data presented in this study are available on request from the corresponding author. The data are not publicly available to protect the privacy of the subjects.

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
