# Peer review of "Next-Generation Sequencing for Determining the Effect of Arginine on Human Dental Biofilms Using an In Situ Model"

_pharmacy, 2021, doi:10.3390/pharmacy9010018_

Round 1

Reviewer 1 Report

General comments

This study presents few significant results. That’s all right: in science it is useful to also describe the negative results. An 8% arginine solution raised the concentration of NH4+ ions in vitro and in vivo in saliva, and decreased the proportion of only two bacterial species, which are not major species in pathogenic oral biofilms. Other microbiologic results are negative or only trends. This is honestly recognized in the Discussion and Conclusion sections, but not in the Abstract.

The review of the literature on oral biofilms is superficial. The characteristics of the biofilm compatible with oral health and the biofilm associated with caries and periodontitis are well-known concepts: detail and cite references.

There are untruths about oral hygiene standards and antiseptics/antibiotic use in dentistry.

Abstract

Clearly mention that an 8% arginine solution raised the concentration of NH4+ ions in vitro and in vivo in saliva, and decreased the proportion of Atopobium and Catonella species in vivo. Atopobium and Catonella are not major bacteria in oral biofilms. Other microbiologic results are negative or only trends.

  1. Introduction

The review of the literature on oral biofilms is superficial: endogenous fungi and viruses should be mentioned, as well as parasites in periodontal pockets. This § must be rewritten, with references. The eventual impact of arginine on non-bacterial components in oral biofilm must be discussed in the Discussion section.

Line 50
“We confirmed” instead of “we found” and add references.

Line 53
At high risk of what? A higher risk of dental caries and periodontitis is not limited to immunocompromised patients. Please detail the major risks for dental caries (sugar, acid, snacking and soda, alcohol, dry mouth, etc.) and periodontitis (tobacco, diabetes, etc.) in addition to poor oral hygiene and insufficient or no follow-up by a dental surgeon. Add references.

Line 54
Suppress “Several approaches…and quorum-sensing related substances”. The control of the oral biofilm is based on a correct lifestyle: oral hygiene with mechanical means (brushing, flossing) and fluoridated toothpaste, lifestyle hygiene with a noncariogenic diet, elimination of toxic habits (alcohol, tobacco, cannabis, other addictive substances), dental follow-up and scaling on a regular basis, etc. Clarify:” antibacterial agents” mean antiseptics and antibiotics. Biofilm control is not based on antibacterial agents, excepted for specific periodontal conditions, to be detailed. Mention quorum-sensing related substances, but not as if they were routine products for oral hygiene. Add references.

Line 57

Suppress “An approach for converting…attention worldwide” in the Introduction and in the Discussion section.

  1. Materials and Methods
  • 2.1.
    Please detail culture media, incubation duration, temperature, aerobic/anaerobic conditions, etc.

Line 90-91
Luciferase enzyme et purified luciferin: detail Company, City, Country.

Line 96
§ 2.2. with a corresponding subtitle is missing.

Line117
Please detail the method used to collect saliva samples.

Figure1
The Figure is unclear or seems uncomplete. Subjects in both Control group and Arginine group benefited from an initial sampling followed by appliance installation at baseline (D0), and from appliance installation followed by sampling at the end of the study (D28). I recommend to suppress Figure 1, and to describe study design in the text.

Line 148
Correct comma position: “,IBM”

  1. Results

Line170
§ 3.2. with a corresponding subtitle is missing.

Line188-189
“The proportion of the Firmicutes phylum increased, and the proportion of the Bacteroidetes phylum decreased”: it must be mentioned that it is just a trend, without statistic validation.

Line 190
“The proportion of Streptococcus and Neisseria”: same remark.

Figure 8 and 9 legends
“Circles represent outliers”: where are these circles?

  1. Discussion

Line 231-232
Suppress “In general…for treating infectious diseases”. This is an untruth concerning oral microbial diseases: dental caries have never been treated with antibiotics, and antibiotic treatment of periodontitis is limited to some severe and specific forms.
Suppress “however the bacteria…resistance to the antibiotics”. Another untruth: antibiotic resistance is not a problem in treating dental caries and in well conducted periodontal treatment.

Line 236
Discuss the impact of increasing the pH on calculous formation and periodontal disease. Add references.
Discuss the importance to limit sugar and acid in food and beverage to control saliva pH.

Line 265-267
Suppress “The genera Atopobium…caries and periodontitis”. Untruth or exaggerated. This part of the Discussion section must be rewritten.

Clearly mention that there is no difference between Control group and Arginine group, as regards the most frequently detected taxa: Atopobium and Catonella count for less than 1% relative abundance.

Atopobium and Catonella are not major pathogens in oral biofilms. Reviewer’s Pubmed research on a 10-year periods:
- Atopobium was associated to vaginosis in 14/15 review articles but not to caries or periodontitis (0/15 review articles).
- Catanella: Alexandra catanella is a dinoflagellate (1/1 review article).

Reference 27, Wu et al., 2018:
In oral biofilms, Catanella is not associated to caries or to periodontitis, only to obesity. Please cite other references.

Reference 28 Aas et al., 2008: 
« Bacterial species other than S. mutans, e.g., species of the genera Veillonella,  Lactobacillus, Bifidobacterium, and Propionibacterium, low-pH non-S. mutans streptococci, Actinomyces spp., and Atopobium spp., likely play important roles in caries progression.” To be discussed.

Author Response

Prof. Dr. Keith A. Wilson

Editor-in-Chief

Pharmacy

Dear Editor:

We wish to submit our revised manuscript entitled “Next-generation sequencing for determining the effect of arginine on human dental biofilms using an in situ model (pharmacy-1013206)” for publication in Pharmacy.

Response to Reviewer Comments

We are grateful to the reviewers for their comments and suggestions, which have helped us to improve our manuscript. We have revised the manuscript based on their comments and have provided our point-by-point responses to each of their comments below.

Point 1: Abstract

Clearly mention that an 8% arginine solution raised the concentration of NH4+ions in vitro and in vivo in saliva, and decreased the proportion of Atopobium and Catonella species in vivo. Atopobium and Catonella are not major bacteria in oral biofilms. Other microbiologic results are negative or only trends.

→ We have modified the text to indicate this.

Lines 22–26: We found that 8% arginine solution significantly increased the concentration of ammonium ions (NH4+) in vitro and in vivo in saliva (P < 0.05) and decreased the proportions of the genera Atopobium and Catonella in vivo. However, the viable count was unaffected by the mouthwash. Furthermore, the oral populations of the genera Streptococcus and Neisseria tended to increase with the use of arginine.

Point 2: Relationship between biofilm and caries / periodontal disease

The review of the literature on oral biofilms is superficial. The characteristics of the biofilm compatible with oral health and the biofilm associated with caries and periodontitis are well-known concepts: detail and cite references.

Introduction

The review of the literature on oral biofilms is superficial: endogenous fungi and viruses should be mentioned, as well as parasites in periodontal pockets. This § must be rewritten, with references. The eventual impact of arginine on non-bacterial components in oral biofilm must be discussed in the Discussion section.

Discussion

Line 231-232 Suppress “In general…for treating infectious diseases”.

Suppress “however the bacteria…resistance to the antibiotics”.

→ The relationship between biofilms and caries/periodontal disease has been summarized in the Introduction and Discussion section as detailed below.

Line 41-46: Not only bacteria but also fungi, viruses, and parasites are present in biofilms. The genus Candida is often detected as a fungus in the oral cavity, and it has been reported that its prevalence is increasing especially in denture wearers and the elderly, and this can lead to invasive infections that have a high mortality rate [3]. In addition, periodontal pockets are rich in nutrients, and parasites such as Trichomonas have been detected, which exacerbate damage to the oral mucosa and have been detected in oral ulcers in kidney transplant patients [4].

Line 46-56: Oral biofilms are formed in an ordered way and retain a diverse microbial composition that remains relatively stable over time in good health [5]. However, imbalances in the oral bacterial flora (dysbiosis) caused by specific stress factors, such as carbohydrate consumption and plaque accumulation, are thought to lead to the development of oral diseases such as caries and periodontal disease [6, 7]. In caries, there is a shift to community domination by acid-producing and acid-resistant species such as Streptococcus mutans and Lactobacillus [8]. Periodontal disease is thought to require colonization by certain pathogens such as Porphyromonas gingivalis [9]. Among them, it is considered that the synergistic and antagonistic actions among the bacterial species in the biofilm enhance or inhibit the colonization and pathogenicity of each [10, 11]. Thus, oral biofilms might lead to the development of oral diseases, and the interactions among its constituent bacteria affect bacterial virulence [12].

Line 256-262: Dental diseases, such as caries and periodontal disease, are currently considered to result from an imbalance (dysbiosis) in the normally stable resident oral flora [35]. This is because perturbations in the microbiome caused by certain stress factors, such as carbohydrate consumption or plaque accumulation, can lead to the development of oral diseases, such as caries or periodontal diseases [5]. Therefore, the approach of maintaining the balance of the bacterial flora by changing the pathogenic bacterial flora to non-pathogenic flora rather than killing the bacteria has attracted attention worldwide [5, 36].

Line 300-303: Bacteria in biofilms also communicate via quorum sensing using small diffusive signaling (CSP), which contributes to an increase in antimicrobial resistance. Thus, the oral bacteria in the biofilm do not exist as independent entities but function as a metabolically integrated microbial community [8]. Therefore, we focused on bacterial metabolism and analyzed functional predictors.

Point 3:

Line 50 “We confirmed” instead of “we found” and add references

→ In our previous study (Reference 5), we found that the number of viable bacteria in supragingival biofilms increase in two steps, and thus, we have revised the text as follows:

Line 64-65: and confirmed that the number of viable bacteria in supragingival biofilms increased in two steps [14].

Point 4:

Line 53 At high risk of what? A higher risk of dental caries and periodontitis is not limited to immunocompromised patients. Please detail the major risks for dental caries (sugar, acid, snacking and soda, alcohol, dry mouth, etc.) and periodontitis (tobacco, diabetes, etc.) in addition to poor oral hygiene and insufficient or no follow-up by a dental surgeon. Add references.

→We have modified the text according to your suggestion.

Line 71-76: In recent years, the number of elderly people has been increasing worldwide, and the risk of developing caries and periodontal disease is increasing. This is due to the deterioration of oral hygiene and the decrease in saliva, which causes acidification of the oral environment and the accumulation of plaque, resulting in imbalances of the oral bacterial flora [6, 7]. In addition, the interrelationship between systemic diseases such as diabetes and heart disease and periodontal disease has been reported [17]. Therefore, the importance of controlling oral biofilms is increasing.

Point 5:

Line 54 Suppress “Several approaches…and quorum-sensing related substances”.

→ As you indicated, no antibacterial or quorum-sensing related substances are used on a daily basis for biofilm control. We have indicated that such control methods are being considered via in vitro studies.

Line 77: Several in vitro studies have been conducted…

Point 6:

Line 57 Suppress “An approach for converting…attention worldwide” in the Introduction and in the Discussion section.

→ We have modified this on lines 260‒262 accordingly.

Line 260-262: Therefore, the approach of maintaining the balance of the bacterial flora by changing the pathogenic bacterial flora to non-pathogenic flora rather than killing the bacteria has attracted attention worldwide [5, 36].

Point 7: Materials and Methods

2.1.Please detail culture media, incubation duration, temperature, aerobic/anaerobic conditions, etc.

→The above contents have been discribed in Lines 105‒106.

Line 90-91

Luciferase enzyme et purified luciferin: detail Company, City, Country.

Line117 Please detail the method used to collect saliva samples.

Line 148 Correct comma position: “,IBM”

The Figure 1 is unclear or seems uncomplete.

→ We have modified the text and Figure 1 according to these points.

Line 112: luciferase enzyme (CellTiter-Glo® Luminescent Cell Viability Assay, Promega, Tokyo, Japan)

Lines 141‒142: Resting saliva was collected, and 10 μL was used for the experiment.

Lines 171‒172: SPSS® Statistics software (version 22.0, IBM SPSS Inc., Chicago, IL, USA)

Point 8: Results

Line188-189

“The proportion of the Firmicutes phylum increased, and the proportion of the Bacteroidetes phylum decreased”: it must be mentioned that it is just a trend, without statistic validation.

Line 190

“The proportion of Streptococcus and Neisseria”: same remark.

→ We have modified the text and Figure 1 accordingly.

Lines 212‒215: Firmicutes tended to increase and the phylum Bacteroidetes tended to decrease in the arginine group compared with levels in the control group (Fig. 6). At the genus level, the genera Streptococcus and Neisseria of the arginine group showed an increasing tendency (Fig. 7).

Figure 8 and 9 legends

“Circles represent outliers”: where are these circles?

→ In Figure 9, there is no circle, and thus, we have deleted this text.

Point 9

Line 236 Discuss the importance to limit sugar and acid in food and beverage to control saliva pH.

→ We have added the following sentences to indicate this.

Lines 264‒266: It is generally advocated that it is important to limit sugars and acids in foods and beverages to control saliva pH [6, 7].

Point 10

Line 265-267 Suppress “The genera Atopobium…caries and periodontitis”. Untruth or exaggerated. This part of the Discussion section must be rewritten. Clearly mention that there is no difference between Control group and Arginine group, as regards the most frequently detected taxa: Atopobium and Catonella count for less than 1% relative abundance.

→ Although Atopobium and Catonella were rarely detected in our study, recent molecular studies have reported that Veillonella, Propionibacterium, and Atopobium have also been added to the list of cariogenic bacteria [39, 40]. However, these are bacteria that have recently been detected and have hardly been studied, and thus, it is considered that further research is required on these bacteria.

The proportion of Porphyromonas gingivalis in patients with periodontal disease is as low as 0.926% when analyzed by NGS 2. However, there is a strong correlation between P. gingivalis and periodontal disease 3, which has been reported as a "keystone" biofilm species in the regulation of host responses 4. In other words, in our study, it is important that the addition of arginine caused changes in the oral biofilm bacterial flora, suggesting that Atopobium and Catenella are keystone species in the oral bacterial flora.  

Accordingly, we have modified the text as below:

Lines 289‒290: This study found that the arginine group tended to have a higher proportion of streptococci and Neisseria, similar to the observation in the bacterial flora of early biofilms.

Lines 293‒299: Atopobium has been added to the list of cariogenic bacteria [39, 40]. Moreover, Atopobium contains species that are rare opportunistic pathogens, even though they have been detected in skin and soft tissue infections, as well as in infective endocarditis in patients with untreated diabetes [41]. Therefore, it is suggested that arginine could have a function in preventing dysbiosis by inhibiting the growth of opportunistic pathogens, such as Atopobium and Catonella. However, these bacteria have been recently detected and have hardly been studied, and thus, further knowledge of these bacteria is required.

References

1: Lang, N.P., et al. Toothbrushing frequency as it relates to plaque development and gingival health J. Periodontol. 1973, 44, 396–405.

2: López-Martínez, J., et al. Bacteria associated with periodontal disease are also increased in health. Med. Oral Patol. Oral Cir. Bucal. 2020, 25, e745-e751. DOI: 10.4317/medoral.23766.

3: Rafiei, M., et al. Study of Porphyromonas gingivalis in periodontal diseases: A systematic review and meta-analysis. Med. J. Islam Repub. Iran. 2017, 31, 62. DOI: 10.18869/mjiri.31.62. eCollection 2017.

4: Bostanci, N., et al. Porphyromonas gingivalis: an invasive and evasive opportunistic oral pathogen. FEMS Microbiol. Lett. 2012, 331, 1–9‏

Reviewer 2 Report

I think that we can approve that article. The topics is of interest in our days when we try not to use too much antibiotics to prevent different inflammatory pathology into the mouth. In our days, there are more and more patients with a fragile status and thus it is of paramount importance to use methods as simple as possible to improve general status.

Author Response

Thank you for your comment.
We will amend our manuscript following another reviewers comment and suggestion.

Thank you again for spending your precious time with me.

Reviewer 3 Report

Dear authors,

Thank you for submitting your manuscript!

The aim of this study was to assess the effect of different arginine preparations on dental biofilm which was growing in in situ model. The overall objective is good. The manuscript needs minor adjustments.

General Comments:

This is a well-structured and a well-written paper! Thank you for the authors for their efforts!

Introduction:

  1. Add a paragraph in the introduction talking about the commonly used preparations to prevent the growth of mature biofilm.

Materials and Methods:

  1. Please add references to this section as appropriate.
  2. Please specify why these specific time-points were selected.

Results:

  1. Please clarify if each experiment was done once or repeated and with how many samples in each?
  2. Please specify the p-value under/in each figure.
  3. Organize the abbreviations under each figure.

Discussion:

  1. In line 247, would you please discuss why do you think there was no significant difference?
  2. Please add a section where you can discuss the differences between early and mature biofilm. Also, discuss the time needed for mature biofilm to develop.
  3. Please check the minor English language errors.

Conclusion:

  1. In line 289, had you measured the pH?
  2. Move the limitation and future researches to the last part of the discussion.

  • Remove lines 300 and 301

Thank you!

Author Response

We are grateful to the reviewers for their comments and suggestions, which have helped us to improve our manuscript. We have revised the manuscript based on their comments and have provided our point-by-point responses to each of their comments below.

Reply to the Review Report 

Add a paragraph in the introduction talking about the commonly used preparations to prevent the growth of mature biofilm.

→We have modified the text to indicate this.

Lines 68-72: indicating that the proportion of anaerobic bacteria that cause gingivitis, such as Fusobacterium, Prevotella, and Porphyromonas [15, 16], increased after 48 h, suggesting a shift to mature biofilms. Therefore, to suppress the growth of such mature biofilms, their mechanical and chemical removal with brushing and mouthwashes are routinely performed. However, a scientifically based method for controlling dental biofilms has not yet been established.

Materials and Methods:

Please specify why these specific time-points were selected.

→ When the effect of sleep on dental biofilm was searched using an in situ model, the result was that the bacterial flora changed immediately after waking up. Therefore, in order to investigate the effect of arginine on the bacterial flora immediately after waking up, this schedule was used.

Results:

Please clarify if each experiment was done once or repeated and with how many samples in each?

Please specify the p-value under/in each figure.

Organize the abbreviations under each figure.

→ We have modified the text under each figure.

Each sample was tested in triplicate.

Discussion:

In line 247, would you please discuss why do you think there was no significant difference?

→ We have added the text.

Lines 283-285: This is because arginine has the ability of altering the oral microbial community from pathogenic bacteria-dominated flora that cause dental caries and periodontal disease to non-pathogenic bacterial flora, not killing the bacteria [21].

Please add a section where you can discuss the differences between early and mature biofilm. Also, discuss the time needed for mature biofilm to develop.

→ We have added the text.

Lines 294-298: A study that investigated the effect of the time interval between brushing one’s teeth and the induction of gum inflammation found that brushing every 48 h increases plaque score formation, although gingivitis did not develop. The authors speculated that these changes to this mature biofilm may be explained by the quantitative and qualitative changes in biofilms that occur approximately 48 h after initial growth [39].

Conclusion:

In line 289, had you measured the pH?

→ The following sentences have been deleted.

Line336: thereby increasing the pH,

Move the limitation and future researches to the last part of the discussion.

→ We have modified the text.

Lines 324-330: In this study, it was predicted that the number of indigenous bacteria in the oral cavity that are not acid-resistant and do not possess arginine-related functional factors would increase in the 8% arginine group. However, in this study, we found that these bacteria and functional factors did not increase in number. In the future, it will be necessary to identify more detailed taxonomic compositions and functional factors using whole genome shotgun sequencing and RNA-seq. Moreover, we intend to develop further dental biofilm control/suppression methods by creating effective arginine preparations, such as adding drugs that enhance the effects of arginine.

Remove lines 300 and 301

→ The above line have been deleted.

Round 2

Reviewer 1 Report

The authors have made the requested changes. The article is gaining clarity and scientific relevance. This revised version may be published without further modification.